# Genome-wide specificity of dCpf1 cytidine base editors

Daesik Kim [1,2,4✉], Kayeong Lim [2,3,4], Da-eun Kim[2,3] & Jin-Soo Kim [2,3✉]

Cpf1-linked base editors broaden the targeting scope of programmable cytidine deaminases by recognizing thymidine-rich protospacer-adjacent motifs (PAM) without inducing DNA double-strand breaks (DSBs). Here we present an unbiased in vitro method for identifying genome-wide off-target sites of Cpf1 base editors via whole genome sequencing. First, we treat human genomic DNA with dLbCpf1-BE ribonucleoprotein (RNP) complexes, which convert C-to-U at on-target and off-target sites and, then, with a mixture of *E. coli* uracil DNA glycosylase (UDG) and DNA glycosylase-lyase Endonuclease VIII, which removes uracil and produces single-strand breaks (SSBs) in vitro. Whole-genome sequencing of the resulting digested genome (Digenome-seq) reveals that, on average, dLbCpf1-BE induces 12 SSBs in vitro per crRNA in the human genome. Off-target sites with an editing frequency as low as 0.1% are successfully identified by this modified Digenome-seq method, demonstrating its high sensitivity. dLbCpf1-BEs and LbCpf1 nucleases often recognize different off-target sites, calling for independent analysis of each tool.

[1] Genome Editing Research Center, Korea Research Institute of Bioscience and Biotechnology (KRIBB), Daejeon 34141, Republic of Korea. [2] Center for Genome Engineering, Institute for Basic Science (IBS), Daejeon 34126, Republic of Korea. [3] Department of Chemistry, Seoul National University, Seoul 08826, Republic of Korea. [4] These authors contributed equally: Daesik Kim, Kayeong Lim. ✉email: dskim89@kribb.re.kr; jskim01@snu.ac.kr

CRISPR RNA-guided programmable deaminases[1–6] [a.k.a. cytidine and adenine base editors (CBEs and ABEs)] comprise (i) a DNA-binding module made up of catalytically deficient Cas9 or Cpf1 and (ii) an engineered cytidine or adenine deaminase. These enzymes respectively convert C-to-T or A-to-G within a single-strand, nontarget DNA bubble generated by the hybridization of the target DNA strand with a guide RNA without inducing DNA double-strand breaks (DSBs). Base editors have been widely used to correct point mutations or induce single nucleotide conversions in eukaryotic cells and whole organisms[7–12].

Catalytically dead Lachnospiraceae bacterium Cpf1 (dLbCpf1; a.k.a. dLbCas12a)-BE was recently developed by linking dLbCpf1 with the cytidine deaminase APOBEC1[13]. Whereas base editor 3 (BE3), which is composed of D10A SpCas9 nickase and APOBEC1, recognizes NGG protospacer-adjacent motif (PAM) sequences and induces C-to-T conversions within positions 4–8 (numbering in the protospacer from 1 to 20 in the 5′–3′ direction)[2], dLbCpf1-BE recognizes TTTV PAM sequences and catalyzes C-to-T conversions within positions 8–13 (numbering in the protospacer from 1 to 23 in the 5′–3′ direction)[13].

Recently, we modified Digenome-seq, originally developed to assess the genome-wide specificities of Cas9[14] and Cpf1[15] nucleases in the human genome, so that it could likewise be used to evaluate the specificities of CBE (BE3)[16] and ABE (ABE 7.10)[17]. For these latter applications, DSBs were induced at uracil- or inosine-containing sites using DNA-modifying enzymes such as Endonuclease VIII or V. In this study, we again modify Digenome-seq to identify SSBs, creat at uracil-containing sites via dLbCpf1-BE, in the human genome and to profile genome-wide specificities of dLbCpf1-BE in human cells. Whole-genome sequencing (WGS) of the resulting digested genome (Digenome-seq) reveals that, on average, dLbCpf1-BE induces 12 SSBs in vitro per crRNA in the human genome. Off-target sites with an editing frequency as low as 0.1% are successfully identified by this modified Digenome-seq method, demonstrating its high sensitivity. dLbCpf1-BEs and LbCpf1 nucleases often recognize different off-target sites, calling for independent analysis of each tool.

## Results

**On-target activity of dLbCpf1-BE and LbCpf1.** We first compared the insertion and deletion (indel) frequencies of LbCpf1 nuclease and the base editing frequency of dLbCpf1-BE at nine human genomic target sites (CDKN2A, RUNX, FANCF, EMX1, DNMT1, LINC01551, DYRK1A, BCL2L13, and CLIC4) in HEK293T cells using targeted deep sequencing. We found that LbCpf1 nuclease (with indel frequences of 47% ± 5%) was more efficient than dLbCpf1-BE (with base editing frequencies of 16% ± 5%) (Supplementary Fig. 1a, b). LbCpf1-induced indel frequencies correlated poorly ($R^2 = 0.56$) with dLbCpf1-BE-induced base editing frequencies. Thus, at some target sites, such as EMX1 and CLIC4, dLbCpf1-BE exhibited low activity, whereas LbCpf1 was highly active (Supplementary Fig. 1a, c). These results show that the dLbCpf1-BE activity can be independent of the LbCpf1 nuclease activity.

**Mismatch tolerance of dLbCpf1-BE and LbCpf1.** Next, we examined the tolerance of dLbCpf1-BE and LbCpf1 for mismatches in crRNAs targeting endogenous genomic loci (Fig. 1; Supplementary Fig. 2). To this end, we treated HEK293T cells with dLbCpf1-BE or LbCpf1 together with a corresponding crRNA containing zero to four mismatches with the target site and measured the resulting frequencies of single-nucleotide substitutions or small indels, respectively, via targeted deep

sequencing. dLbCpf1-BE and LbCpf1 nucleases tolerated most one- and two-base mismatches in the PAM-distal region, but did not tolerate two- to four-base mismatches in the PAM proximal region. Although dLbCpf1-BE is derived from LbCpf1, at some sites, such as those indicated by asterisks in Fig. 1 and Supplementary Fig. 2, the tolerance of dLbCpf1-BE and LbCpf1 nuclease for mismatched crRNAs differed. For example, the relative editing frequency (the editing frequency with the mismatched crRNA divided by that with the matched crRNA) of dLbCpf1-BE complexed with a crRNA containing three mismatches (at positions 19–21 in the protospacer) at the CDKN2A site was 49%, indicating a high tolerance for mismatches, whereas the relative frequency of editing by LbCpf1 nuclease was just 2%, indicating poor activity in the presence of mismatches. In contrast, the relative frequency of editing by dLbCpf1-BE complexed with a crRNA containing mismatches at positions 7 and 8 in the CDKN2A site was only 9%, whereas that of LbCpf1 nuclease was 41%. Given these results, we anticipated that dLbCpf1-BE and LbCpf1 nuclease would cause different off-target effects in the human genome. Therefore, a method was required to evaluate the genome-wide specificity of dLbCpf1-BE in an unbiased manner.

**Identification of genome-wide dLbCpf1-BE off-target sites.** In our previous study[16,17], we profiled the genome-wide off-target effects of BE3 and ABE 7.10 using Digenome-seq. In these experiments, we respectively generated DNA DSBs via treatment with BE3 and USER, a mixture of Escherichia coli uracil DNA glycosylase (UDG) and DNA glycosylase-lyase Endonuclease VIII that is used to remove uracil, or ABE 7.10 and Endonuclease VIII (Endo VIII). Note, however, that treatment with dLbCpf1-BE and USER will not induce DNA DSBs at target loci, but instead leads to DNA SSBs. This occurs because, unlike BE3, which comprises a Cas9 nickase and a cytidine deaminase, dLbCpf1-BE consists of a catalytically dead dLbCpf1 and a cytidine deaminase.

As a first step of modifying Digenome-seq for use with dLbCpf1-BE, we tested whether DNA SSBs generated by dLbCpf1-BE and USER could be detected by WGS. Genomic DNA purified from HEK293T cells was first incubated with RNP complexes containing purified dLbCpf1-BE protein and an in vitro transcribed crRNA targeting DYRK1A to induce C-to-U conversions at the target locus, and then with USER to remove dLbCpf1-BE-generated uracil (Fig. 2a). To confirm that these events occurred, we amplified DNA from the on-target regions of untreated, dLbCpf1-BE-treated, and dLbCpf1-BE- and USER-treated samples, and performed Sanger sequencing (Fig. 2b). Note that polymerase chain reaction (PCR) amplification will change dLbCpf1-BE-generated uracils to thymines. Positions at which uracils were removed by USER revert to cytosine in the Sanger sequencing results, because the digested Watson strands cannot be amplified, but the undigested Crick strands can be amplified (Fig. 2b).

After confirmation that SSBs were induced by treatment with dLbCpf1-BE and USER, genomic DNA was subjected to WGS after fragmentation, end repair, and adapter ligation. Sequence reads were aligned to the human reference genome (hg19). Using Integrative Genomics Viewer (IGV), we then observed sequence reads aligned to the reference genome at the on-target site. Interestingly, Watson strands, digested by USER, and Crick strands, undigested, represent straight and staggered alignments, respectively (Fig. 2c). To evaluate potential dLbCpf1-BE off-target sites in the human genome, we divided the sequence reads into forward and reverse strands and assigned the number of sequence reads with 5′ ends that started at a given position to each nucleotide (nt) position across the genome via SAMtools. Then, we computationally captured the sites identified by modified

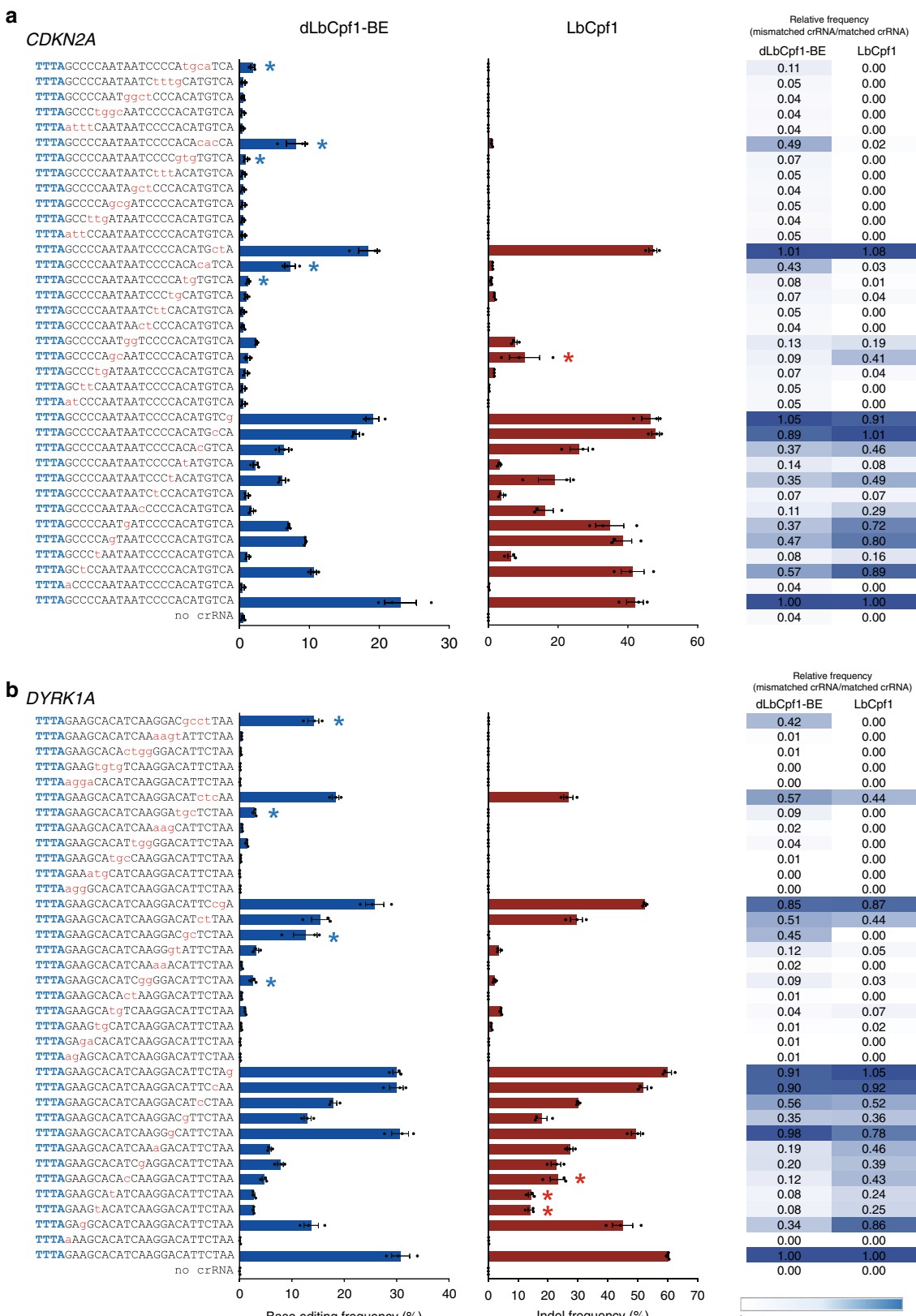

Digenome-seq that satisfied the following requirements: (i) the count of sequence reads with the same 5′ end was greater than 5 and at least 20% of the sequence reads exhibited a straight alignment at a given position and (ii) that contained a PAM sequence (5′-TTTN-3′) and had 8 or fewer mismatches with the target sequence or contained PAM-like sequences (5′-NTTN-3′,

5′-TNTN-3′, or 5′-TTNN-3′) and had 7 or fewer mismatches with the target sequence (Fig. 2d). Note that the mismatches at positions 21–23 in the protospacer had little effect on editing efficiency (Fig. 1; Supplementary Fig. 2), so we used a 20-nt protospacer sequence to count the number of mismatches by comparing it with SSB sites and on-target sequence. We chose the

**Fig. 1 Tolerance of dLbCpf1-BE and LbCpf1 for mismatched crRNAs. a, b** Plasmids encoding matched and mismatched crRNAs (containing 1- to 4-nt mismatches relative to the on-target site) targeting sites in *CDKN2A* (**a**) and *DYRK1A* (**b**) were transfected into HEK293T cells together with plasmids encoding dLbCpf1-BE or LbCpf1. Base editing and indel frequencies were measured using targeted deep sequencing. Mismatched bases and PAM sequences are shown in red and blue, respectively. The relative activity (the editing frequency with the mismatched crRNA divided by that with the matched crRNA) indicates the mismatch tolerance. Mismatched crRNAs for which the relative activity of dLbCpf1-BE is at least three times higher than that of LbCpf1 are indicated with blue asterisks; those for which the relative activity of LbCpf1 is at least three times higher are indicated with red asterisks. Data are shown as mean ± s.e.m. from three biologically independent samples. Source data are provided as a Source Data file.

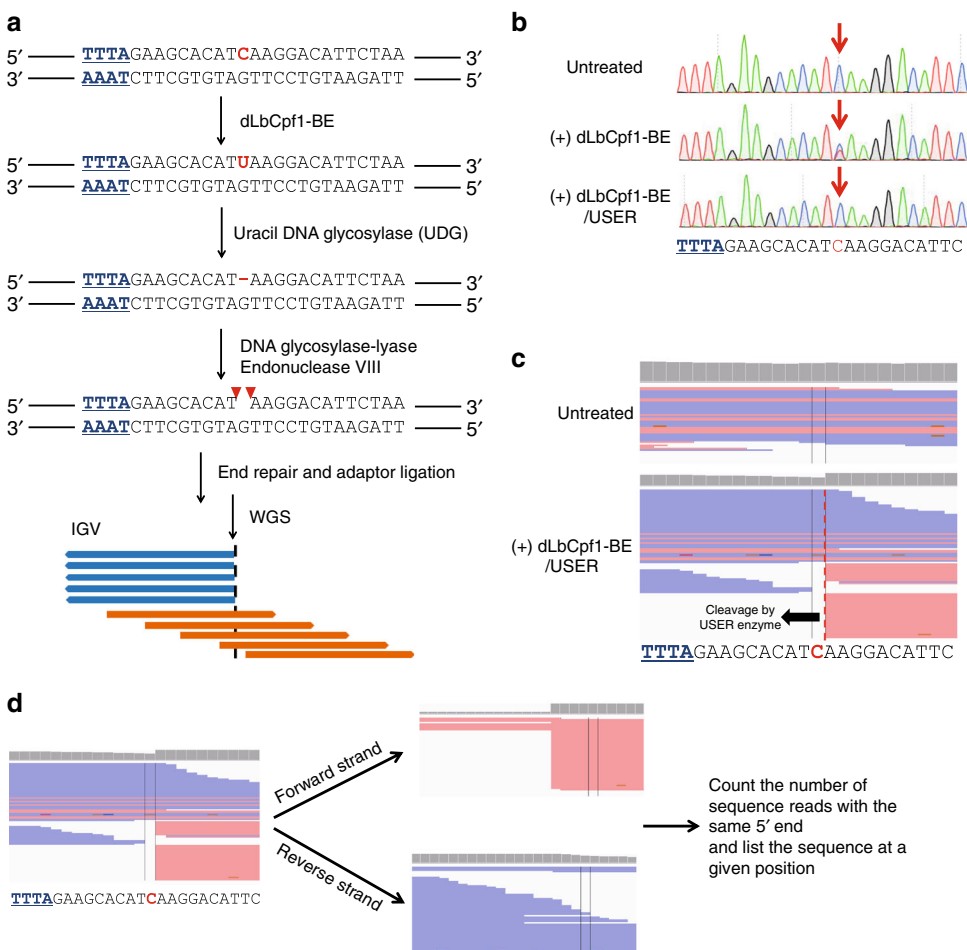

**Fig. 2 Determination of the genome-wide specificity of dLbCpf1-BE using Digenome-seq. a** Overview of dLbCpf1-BE-mediated Digenome-seq. Genomic DNA is incubated with dLbCpf1-BE, which converts C-to-U within the base editing window. A single nucleotide gap is generated by treatment with USER, a mixture of UDG and DNA glycosylase-lyase Endonuclease VIII. UDG recognizes and removes a uracil base to form an abasic site, shown as a red dash. DNA glycosylase-lyase Endonuclease VIII breaks the phosphodiester backbone of the abasic site, forming a DNA SSB as indicated with the red arrowheads. The dLbCpf1-BE- and USER-treated DNA is then subjected to WGS to evaluate the genome-wide specificity of dLbCpf1-BE. **b** Representative Sanger sequencing results showing dLbCpf1-BE-induced C-to-U conversion and USER-induced uracil excision. The position of the sequence change is indicated by the red arrows. **c** IGV images showing straight and staggered alignments of sequence reads at the DYRK1A on-target site obtained using WGS. **d** Workflow of WGS analysis to capture genome-wide SSB sites. Sequence reads are divided into forward and reverse strands according to the read orientation. The number of sequence reads with 5′ ends at each position are then counted for each set of strands to profile off-target sites.

mismatch number of 8 as the cutoff value at PAM (5′-TTTN-3)- containing sites, which is a less strict parameter than that used in other genome-wide Cpf1 off-target profiling methods such as GUIDE-seq (allowing for up to seven mismatches)[18] and BLISS (allowing for up to four mismatches)[19], because WGS data obtained with intact genomic DNA did not show any false-positive sites with eight or fewer mismatches (Supplementary Table 1). Using these bioinformatics analyses, we identified 20 potential off-target sites, in addition to the on-target site, by modified Digenome-seq using dLbCpf1-BE with a crRNA targeting *DYRK1A* (Supplementary Table 2). To check the

reproducibility of Digenome-seq, we independently performed Digenome-seq using the *DYRK1A*-targeted dLbCpf1-BE twice, and found that the same 17 sites including the on-target site were captured by the two different experiments (Supplementary Fig. 3; Supplementary Table 2). These experiments established that we could identify dLbCpf1-BE off-target sites using Digenome-seq with high reproducibility.

We carried out additional modified Digenome-seq experiments using dLbCpf1-BE with crRNAs targeting eight different genomic sites. Between 1 and 46 SSB sites were captured per crRNA in vitro (average, 12 ± 5 SSBs per crRNA; total, 9 target sites)

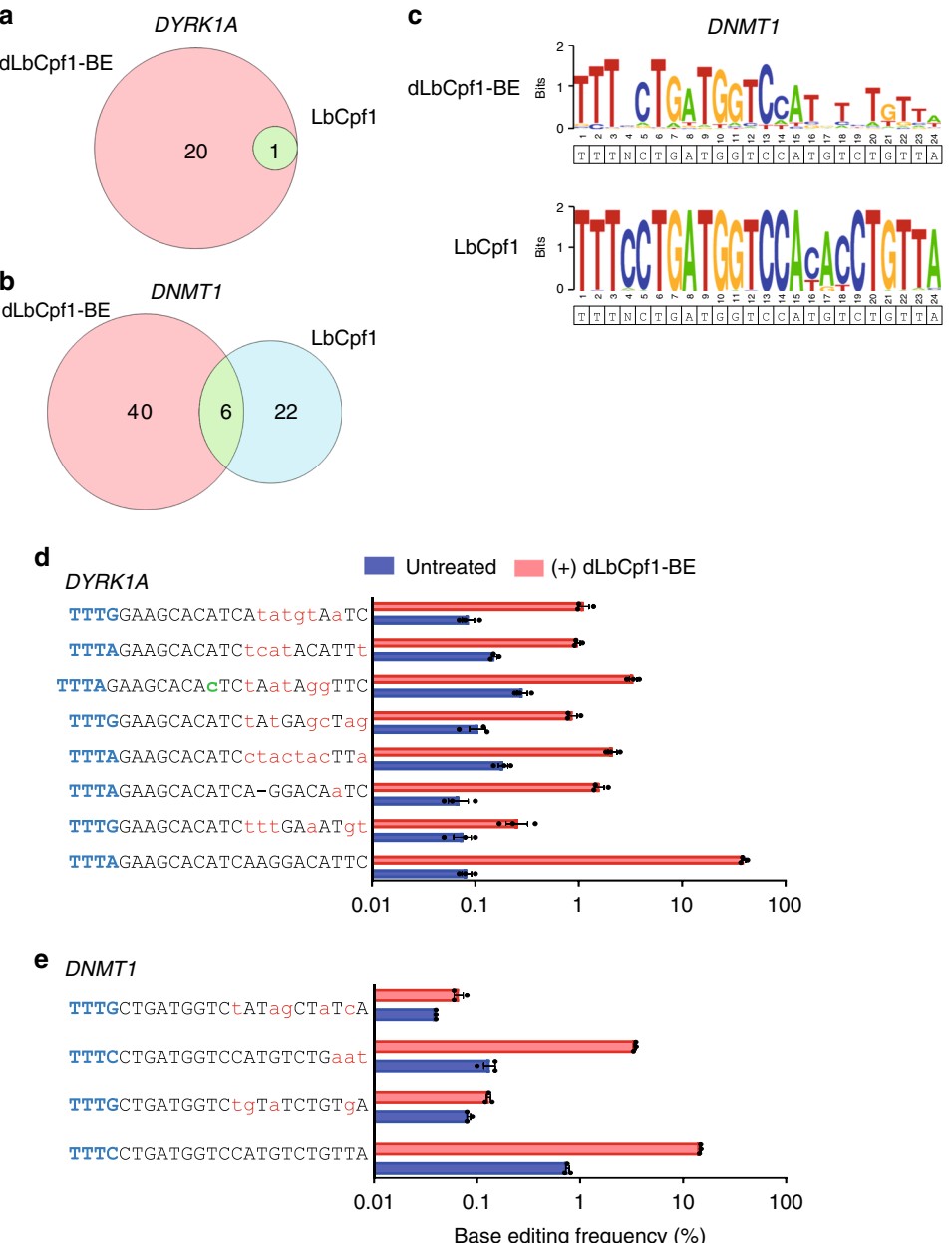

**Fig. 3 Genome-wide off-target sites of dLbCpf1-BE revealed by Digenome-seq and targeted deep sequencing. a, b** Venn diagrams showing the number of in vitro cleavage sites in the human genome produced by dLbCpf1-BE or LbCpf1 nuclease targeting the *DYRK1A* (**a**) and *DNMT1* (**b**) sites that were identified by Digenome-seq. **c** Sequence logos were generated for DNA sequences captured by dLbCpf1-BE- and LbCpf1-mediated Digenome-seq using WebLogo. **d, e** Base editing frequencies of dLbCpf1-BE at in vitro cleavage sites, identified by dLbCpf1-BE-mediated Digenome-seq, were measured using targeted deep sequencing of genomic DNA from HEK293T cells. Mismatched bases and PAM sequences are shown in red and blue, respectively. An RNA bulge is represented as a dash and a DNA bulge is shown in green. Data are shown as mean ± s.e.m. from three biologically independent samples.

including the on-target site (Supplementary Table 3). Note that although a different detection method was used with dLbCpf1-BE, the average number of in vitro LbCpf1 cleavage sites was 6 ± 3[2], which is fewer than that of dLbCpf1-BE.

**Off-target sites of dLbCpf1-BE revealed by Digenome-seq.** We next examined the relationship between the in vitro cleavage sites induced by LbCpf1 nuclease and dLbCpf1-BE when they were targeted to the *DYRK1A* site. Compared to the LbCpf1 nuclease, which produced one cleavage site in vitro at the on-target site, dLbCpf1-BE produced 21 cleavage sites in vitro (Fig. 3a). In addition, we also compared the in vitro cleavage sites induced by

LbCpf1 nuclease and dLbCpf1-BE targeted to three additional target sites (*DNMT1*, *CKDN1A*, and *EMX1*) (Fig. 3b; Supplementary Fig. 4). Only 9 out of 73 (=12%) of the cleavage sites identified using dLbCpf1-BE targeting *DYRKA1A*, *DNMT1*, *CDKN1A*, and *EMX1* were also identified using LbCpf1 nucleases targeting the same sites (Fig. 3a, b; Supplementary Fig. 4). A sequence logo, generated from the DNA sequences at in vitro LbCpf1 nuclease cleavage sites, indicated that all nucleotide positions contributed to specificity (Fig. 3c). However, a sequence logo generated from the dLbCpf1-BE-modified sites showed that the sequence of the PAM distal region was less important than the seed sequence for specificity (Fig. 3c). In addition, whereas 17% (18 of 106) of the dLbCpf1-BE cleavage sites identified by

Digenome-seq have missing or extra nucleotides compared to their respective on-target site, respectively resulting in RNA or DNA bulges (Supplementary Table 2), only 2% (1/49; data from previous study[15]) captured by Digenome-seq using LbCpf1 were associated with an RNA bulge. These results indicate that dLbCpf1-BE and LbCpf1 nuclease recognize different off-target sites.

**Validation of dLbCpf1-BE off-target sites in a human cell**. To validate the off-target sites identified by dLbCpf1-BE-mediated Digenome-seq in vitro, we measured dLbCpf1-BE-mediated substitution frequencies in HEK293T cells using nine different crRNAs. Among 106 candidate sites, including all nine on-target sites, 29 were validated via targeted deep sequencing; these included all of the on-target sites (Fig. 3d, e; Supplementary Table 4). The 20 validated off-target sites exhibited base editing frequencies that ranged from 0.1 to 22.4% (in comparison, on-target editing frequencies ranged from 2.4 to 39.5%) (Supplementary Fig. 5). These findings, which showed that we could identify dLbCpf1-BE off-target sites with base editing frequencies as low as 0.1%, demonstrated that dLbCpf1-BE mediated-Digenome-seq is a highly sensitive method. We also found that the ratio of the count of sequence reads with the same 5′ end to the read depth in Digenome-seq results at validated dLbCpf1-BE off-target sites was poorly correlated with mutation frequencies in HEK293T cells ($R^2 = 0.30$) (Supplementary Fig. 6). At the same time, we also measured LbCpf1 nuclease-mediated indel frequencies at the same 106 candidate sites for comparison (Supplementary Table 4). Against expectations, LbCpf1 showed off-target activity at only 2 of the 20 off-target sites validated for dLbCpf1-BE, even though the nuclease exhibited higher on-target activities (with indel frequencies that ranged from 24.1 to 68.9%) than dLbCpf1-BE (Fig. 3d, e, Supplementary Fig. 5). Collectively, these results reinforced the idea that dLbCpf1-BE and LbCpf1 differ in their off-target activities. We next focused on the off-target sites that were captured by LbCpf1-, but not dLbCpf1-BE-mediated Digenome-seq (Fig. 3a, b; Supplementary Fig. 4). None of these 25 sites were validated by targeted deep sequencing in dLbCpf1-BE-transfected HEK293T cells (Supplementary Table 5).

In addition, we compared the off-target effect (OTI index), the ratio of the sum of the mutation frequencies at validated off-target sites to the mutation frequency at the on-target site, of dLbCpf1-BE and BE3 (APOBEC1–Cas9 D10A nickase–UGI). We found that the OTI of dLbCpf1-BE ($0.27 \pm 0.16$) was less than that of BE3 ($0.45 \pm 0.25$)[16], suggesting that the specificity of dLbCpf1-BE is higher than that of BE3 (Supplementary Table 6).

**Reducing dLbCpf1-BE off-target effects via APOBEC1 variants**. To minimize dLbCpf1-BE off-target activity, we first replaced conventional crRNA (spacer length: 23 nt) with truncated crRNAs (spacer length: 16, 18, or 20 nt) or extended crRNAs (spacer length: 25 or 27 nt) and measured base editing efficiencies in HEK293T cells. These crRNA modifications did not lead to a significant improvement in dLbCpf1-BE specificity (Supplementary Fig. 7). We next incorporated mutations into dLbCpf1-BE that affect amino acid residues in the Cpf1 domain that contact either the target or nontarget DNA to attenuate dLbCpf1-BE activity[20]. Although a K881A mutation in dLbCpf1-BE showed a tendency to improve specificity at several sites, there was no significant overall change associated with any of the mutations, including an N260A mutation in dLbCpf1-BE that corresponds to the mutation in a high fidelity version of AsCpf1 (Supplementary Fig. 8). Finally, we introduced mutations into the cytidine deaminase domain of dLbCpf1-BE, which was known to

narrow the base editing window and increase the specificity of BE3[21], and found that dLbCpf1-BE-YE1 (containing W90Y + R126E mutations in the cytidine deaminase domain) exhibited improved base editing specificity, up to 15-fold better than dLbCpf1-BE, albeit with a lower range of on-target substitution frequencies than dLbCpf1-BE (Fig. 4; Supplementary Fig. 9).

## Discussion

In this study, we modified Digenome-seq to evaluate the genome-wide specificity of dLbCpf1-BE. We successfully identified SSB sites throughout the genome after treating genomic DNA with dLbCpf1-BE and USER in vitro. The average number of potential off-target sites, defined as those sites with a PAM sequence (5′-TTTN-3′) and having eight or fewer mismatches compared to the on-target site, was $2,47,500 \pm 42,200$, making it practically impossible to validate them in cells (Supplementary Table 7). However, on average, dLbCpf1-BE- and USER-mediated Digenome-seq identified 12 in vitro cleavage sites (including the on-target site) per crRNA, a testable number. We confirmed that a subset of these sites were dLbCpf1-BE off-target sites in human cells; we detected sites that were edited at a frequency of at least 0.1%. We also found that dLbCpf1-BE and LbCpf1 nuclease recognize different off-target sites through experiments using mismatched crRNAs, modified Digenome-seq, and targeted deep sequencing. We anticipate that this method will be widely used to assess the genome-wide off-target effects of dCpf1 cytidine base editors. We also improved dLbCpf1-BE specificity by introducing modifications in the cytidine deaminase domain.

In previous studies, base editors consisting of Cas9 nickase and APOBEC1 induced not only gRNA-dependent off-target mutations[16] but also gRNA-independent DNA or RNA mutations[22–24]. Theoretically, dLbCpf1-BE could likewise induce crRNA-independent DNA or RNA editing in cells. To fully understand the off-target effects of dLbCpf1-BE, further studies are needed to identify crRNA-independent off-target effects.

## Methods

**Plasmid construction**. pET-dLbCpf1-BE, a plasmid encoding a human codon-optimized dLbCpf1-BE with a His purification tag at the N terminus (His$_6$-NLS-APOBEC1-XTEN-dLbCpf1(D832A + E925A + D1148A)-NLS-UGI-NLS), was generated using NEBuilder® HiFi DNA Assembly Master Mix (New England Biolabs) to insert dLbCpf1-BE from pCMV-dLbCpf1-BE (Addgene, #107685) into the pET28a vector (Novagen).

pCMV-dLbCpf1-BE (Addgene, #107685) was modified to incorporate mutations in the LbCpf1 domain (N256A, N260A, S348A, K514A, K881A, or K897A; these mutations respectively correspond to N278A, N282A, S376A, K523A, K949A, or K965A in AsCpf1) and combinations of mutations in the APOBEC1 domain (YE1: W90Y + R126E; YE2: W90Y + R132E; EE: R126E + R132E; YEE: W90Y + R126E + R132E) using site-directed mutagenesis (Q5 Site-Directed Mutagenesis Kit, New England Biolabs). crRNA-encoding plasmids were constructed by ligation (Quick Ligation Kit, New England Biolabs) of annealed oligonucleotides to pU6-Lb-crRNA (Addgene, #78957) digested with BsmBI.

**Cell culture and transfection**. HEK293T (ATCC, CRL-11268) cells were cultured in Dulbecco's Modified Eagle's Medium (DMEM) supplemented with 10% (v/v) fetal bovine serum and 1% (v/v) penicillin/streptomycin (Welgene) at 37 °C with 5% CO$_2$. HEK293T cells (~$7.5 \times 10^4$) were seeded on 48-well plates (Corning) and transfected at ~70% confluency with plasmids encoding dLbCpf1-BE (750 ng) and crRNA (250 ng) using 1.5 μL of Lipofectamine 2000 (Invitrogen) according to the manufacturer's protocols.

**Genomic DNA preparation**. Genomic DNA was isolated using a DNeasy Blood & Tissue Kit (Qiagen) at 72 h post transfection. For large-scale analysis, genomic DNA was extracted using 100 μL of cell lysis buffer (50 mM Tris-HCl, pH 8.0 (Sigma-Aldrich), 1 mM EDTA (Sigma-Aldrich), 0.005% sodium dodecyl sulfate (Sigma-Aldrich)) supplemented with 5 μL of Proteinase K (Qiagen). The solution was incubated at 55 °C for 1 h, and then at 95 °C for 10 min.

**Expression and purification of dLbCpf1-BE protein**. BL21 Star (DE3) competent *E. coli* cells (ThermoFisher Scientific) were transformed with the pET-dLbCpf1-BE plasmid, plated on a Luria–Bertani (LB) agar plate containing 50 μg/mL

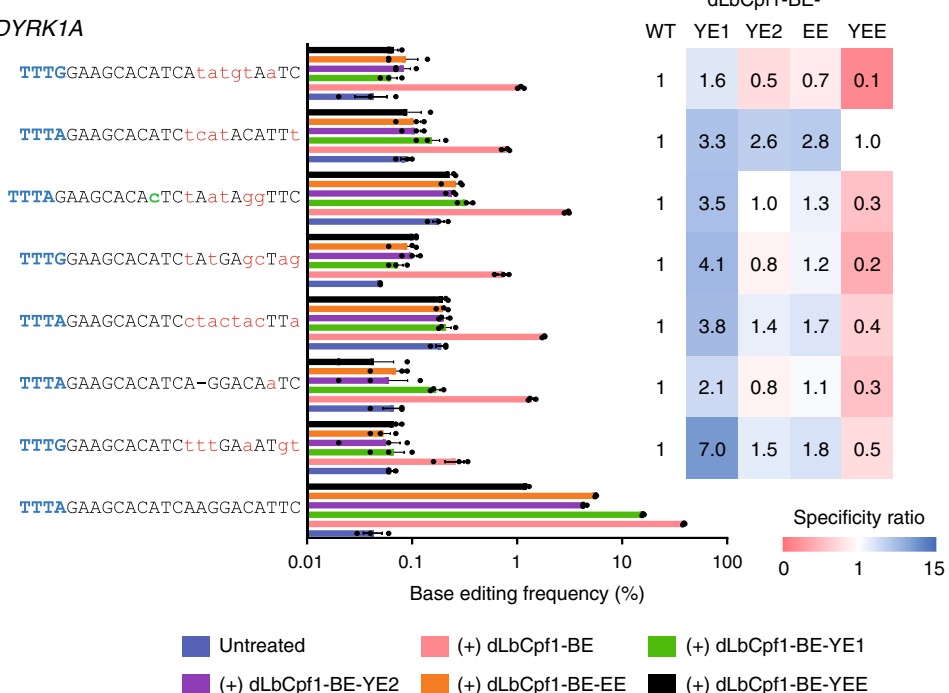

**Fig. 4 Reducing dLbCpf1-BE off-target effects by incorporating mutations in the APOBEC1 domain.** Plasmids encoding versions of dLbCpf1-BE containing mutations in the APOBEC1 domain (YE1: W90Y + R126E; YE2: W90Y + R132E; EE: R126E + R132E; YEE: W90Y + R126E + R132E) were transfected into HEK293T cells to determine if these mutations would improve base editing specificity. Base editing frequencies at on- and -off target sites were determined by targeted deep sequencing. The specificity ratios were calculated by dividing (base editing frequency of dLbCpf1-BE variants at on-target/that at off-target) by (base editing frequency of dLbCpf1-BE-WT at on-target/that at off-target). Mismatched bases and PAM sequences are shown in red and blue, respectively. An RNA bulge is represented as a dash and a DNA bulge is shown in green. Data are shown as mean ± s.e.m. from three biologically independent samples. Source data are provided as a Source Data file.

kanamycin, and incubated overnight at 37 °C. A fresh single colony was selected from the LB agar plate, inoculated into LB medium containing 50 μg/mL kanamycin, and incubated overnight at 37 °C with shaking. The precultures were diluted 1:50 into LB medium supplemented with 50 μg/mL kanamycin and incubated at 37 °C with shaking until the OD600 reached 0.5–0.6. The cultures were cooled on ice, supplemented with 1 mM isopropyl-β-D-1-thiogalactopyranoside (GoldBio), and incubated for ~16 h at 18 °C with shaking.

All subsequent protein purification steps took place at 4 °C. The protein purification buffers, all supplemented with fresh 1 mM dithiothreitol (DTT, GoldBio) and 1 mM phenylmethylsulfonyl fluoride (Sigma-Aldrich), were prepared as follows: Ni-NTA lysis buffer (50 mM sodium phosphate (Sigma-Aldrich), 500 mM NaCl (Sigma-Aldrich), 10 mM imidazole (Sigma-Aldrich), 1% Triton X-100 (Sigma-Aldrich), 20% glycerol, pH 8.0), Ni-NTA wash buffer (50 mM sodium phosphate (Sigma-Aldrich), 150 mM NaCl (Sigma-Aldrich), 35 mM imidazole (Sigma-Aldrich), 20% glycerol, pH 8.0), Ni-NTA elution buffer (50 mM sodium phosphate (Sigma-Aldrich), 150 mM NaCl (Sigma-Aldrich), 250 mM imidazole (Sigma-Aldrich), 20% glycerol, pH 8.0), heparin wash buffer (50 mM sodium phosphate (Sigma-Aldrich), 150 mM NaCl (Sigma-Aldrich), 20% glycerol, pH 8.0), heparin elution buffer (50 mM sodium phosphate (Sigma-Aldrich), 750 mM NaCl (Sigma-Aldrich), 20% glycerol, pH 8.0).

Cells were pelleted by centrifugation at 5000g for 10 min, and re-suspended in 10 mL of Ni-NTA lysis buffer supplemented with 1 mg/mL lysozyme (Sigma-Aldrich) per 800 mL culture. The suspensions were lysed by three repeated freeze (in liquid nitrogen) and thaw (in a water bath) cycles, and sonication with 5 s (on) and 10 s (off) cycles for 9 min. The lysates were cleared by centrifugation at 15,000g for 20 min. Ni-NTA agarose (QIAGEN) was pre-washed with Ni-NTA lysis buffer and incubated with cleared lysates for 60 min while rotating at 4 °C. The mixture was applied to a column and washed two times with Ni-NTA wash buffer, and bound protein was eluted with Ni-NTA elution buffer. Heparin agarose beads (Heparin Sepharose 6 Fast Flow, GE Healthcare) were loaded into a new column and pre-washed with Ni-NTA elution buffer. The eluted protein fraction from the Ni-NTA column was next loaded into the pre-washed heparin column and washed two times with heparin wash buffer. Bound protein was eluted with heparin elution buffer, and eluted protein fractions ware concentrated by centrifugation using Amicon Ultra-4 Centrifugal Filter Devices (Millipore) at 5000g.

**dLbCpf1-BE-mediated in vitro deamination**. Genomic DNA was isolated from HEK293T cells using a DNeasy Blood & Tissue Kit (Qiagen) according to the

manufacturer's instructions. The mixture was treated with RNase A (Qiagen) to remove the residual RNA, after which the DNA was purified again with a DNeasy Blood & Tissue Kit (Qiagen). The in vitro transcribed crRNA (900 nM) was incubated with the purified dLbCpf1-BE protein (300 nM) at room temperature for 10 min. A total of 10 μg of purified genomic DNA was incubated with pre-complexed dLbCpf1-BE RNPs in a reaction volume of 400 μL in reaction buffer (50 mM Tris-HCl (Sigma-Aldrich) (pH 8.0), 25 mM KCl (Sigma-Aldrich), 2.5 mM MgSO₄ (Sigma-Aldrich), 0.1 mM EDTA (Sigma-Aldrich), 10 % glycerol, 2 mM DTT (GoldBio), 10 μM ZnCl₂ (Sigma-Aldrich)) at 37 °C for 8 h. The digested DNA was incubated with RNase A (50 μg/mL, Qiagen) to remove crRNA and then purified with a DNeasy Blood & Tissue Kit (Qiagen). Two microgram of purified DNA was incubated with USER (10 units, New England Biolabs) in a reaction volume of 200 μL at 37 °C for 2 h, and purified again with a DNeasy Blood & Tissue Kit (Qiagen). The target site was amplified by PCR and subjected to Sanger sequencing to check for dLbCpf1-BE-mediated deamination and USER-mediated formation of DNA SSBs.

**WGS and Digenome sequencing**. 1 μg of DNA treated with dLbCpf1-BE and USER in vitro was sheared to a fragment size of 400–500 bp using the Covaris system (Thermo Fisher Scientific). Fragmented DNA was incubated with End Repair Mix (Illumina) and ligated with Illumina-indexing adapters. Sequencing libraries were purified and subjected to WGS using a HiSeq X Ten Sequencer (Illumina) with a sequencing depth of 30–40× at Macrogen. Isaac aligner was used to align sequencing reads to the reference genome sequence. DNA SSB sites were identified using the original Digenome programs. The source code of the original version of Digenome used in this paper is available at https://github.com/snugel/digenome-toolkit.

**Targeted deep sequencing**. On- and off-target sites were amplified from genomic DNA using KAPA HiFi HotStart DNA polymerase (Roche) according to the manufacturer's protocols. The region of interest was first amplified to a size of ~500 bp, after which the amplicons were again amplified to a size of ~200 bp using the primer pairs listed in Supplementary Table 8. PCR amplicons were amplified again using Illumina TruSeq HT dual index primers to label each sample. The PCR products were purified using a PCR purification kit (MGmed). The sequencing libraries were sequenced using MiniSeq (Illumina) with paired-end sequencing systems (2 × 150 bp).

**Statistical analyses**. All results from experiments with three replicates were expressed as mean ± s.e.m.. Comparisons between treated and untreated samples were made using the two-tailed Student's $t$ test. Statistical analysis was performed in Graph Pad PRISM 8.3.1. The colored asterisks in Fig. 1 and Supplementary Fig. 2 were used to indicate differences greater than three-fold in order to highlight dissimilarities in the patterns of dLbCpf1-BE and LbCpf1 activity.

**Reporting summary**. Further information on research design is available in the Nature Research Reporting Summary linked to this article.

## Data availability

Sequencing data have been deposited in the NCBI Sequence Read Archive (SRA) database with BioProject accession code PRJNA630828 and PRJNA634784. Data underlying Figs.1 and 4 and Supplementary Figs. 2, 7, 8, and 9 are provided as a Source Data file. The plasmid encoding dLbCpf1-BE for bacterial expression (pET-dLbCpf1-BE, Addgene, #154256) and plasmids encoding dLbCpf1-BE with APOBEC1 variants for mammalian expression (pCMV-dLbCpf1-BE-YE1, Addgene, #154145; pCMV-dLbCpf1-BE-YE2, Addgene, #154146; pCMV-dLbCpf1-BE-EE, Addgene, #154147; and pCMV-dLbCpf1-BE-YEE, Addgene, #154148) are available from Addgene. Any other additional relevant data are available from the authors upon reasonable request.

## Code availability

The source code of the version of Digenome used in this manuscript is available at https://github.com/snugel/digenome-toolkit.

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

## Acknowledgements

This research was supported by grants from the Institute for Basic Science (IBS-R021-D1) to J.-S.K. and D.K., the KRIBB Research Initiative Program to D.K., and the R&D Convergence Program of the National Research Council of Science & Technology (CAP-15-03-KRIBB) to D.K.

## Author contributions

J.-S.K. supervised the research. J.-S.K., D.K., and K.L. wrote the paper. D.K., K.L., and D.-E.K. performed the experiments and bioinformatics analysis.

## Competing interests

J.-S.K. is a founder of and shareholder in ToolGen, Inc. The remaining authors declare no competing interests.
