## [Peer Review File · Nature Communications]

Reviewers' Comments:

Reviewer #1:

Remarks to the Author:

In the present study, Kim and colleagues intended to assess specificity of dCpf1 base editor, an alternative deamination-mediated base editing system that requires distinct PAM sequence and does not involve DNA nicking. By modifying their Digenome-seq method, which is based on in vitro digestion of isolated genome followed by WGS, the authors tried to provide a comprehensive assay to detect genome-wide deaminated sites. The authors argued that their method is highly sensitive and detects distinct off-target candidates from the ones of the nuclease version. However, as this method seems to detect many false positives and depends on the mismatch criteria to narrow down the number of candidate sites, the reviewer is skeptical of how much their modified Digenome-seq contributes to improve the detection of unexpected off-targets. Besides, the present study cannot address the recent issue of gRNA-independent off-targets for cytosine base editors. The findings that the off-targets for dCpf1 base editor are partly distinct from Cpf1, and that the Apobec1 variants reduces gRNA-dependent off-target, are not quite surprising. Therefore, this reviewer cannot recommend this manuscript to be published in Nature Communications.

Major comments

1. The authors need to mention and estimate the false positive rate of their method and perform comparison with pure prediction-based detection. It will be also beneficial to score off-target candidates based on their Digenome-seq result and compare them with actual off-target rates in the cells.
2. If economically and technically acceptable, false positives could be reduced by increasing the depth of the reads, using other NGS system with longer reads, or enrichment of SSB samples by devising size selection (i.e. producing larger fragments at shearing, followed by end repair and size selection for smaller fragments).
3. The authors need to assess if USER treatment alone may cause significant SSB.
4. The readers would be more interested if the authors could discuss if dCpf1 base editor is superior in its specificity, compared to conventional SpCas9-based one.

Minor comments

L128-129: "A total of 123,309 SSB sites were captured at the DYRK1A site."

Is this supposed to mean in the entire genome? Even so, this huge number suggests that this assay captures too much false positives probably due to technical and statistical limitation. The filtering based on the mismatch criteria cannot solve this statistical problem and thus the false positives remain.

L143-144: "Note that the average number of in vitro LbCpf1 cleavage sites was 6 ± 3 , which is fewer than that of dLbCpf1-BE."

As the detection method for LbCpf1 nuclease off-target is different and perhaps more accurate, it may not be fair to compare these numbers without considering false positives.

L164-165: "The 20 validated off-target sites exhibited base editing frequencies that ranged from 0.1% to 22.4%."

Considering the detection limit of amplicon-seq by the NGS system, is it really possible to detect 0.1% SNV with statistical significance?

Reviewer #2:

Remarks to the Author:

The manuscript "Genome-wide specificity of dCpf1 cytidine base editors" describes off-target activity of dCpf1-BE (C-to-U base editor fused to catalytically inactive dCpf1) in comparison to Cpf1 nuclease. Targeted amplicon sequencing reveals that the dCpf1 base editor is more tolerant to mismatches than the Cpf1 nuclease. The authors describe a variation of Digenome-seq, a method that has been developed in the same lab and used to identify genome-wide off targets of RNA-guided nucleases (Cas9 and Cpf1) as well as Cas9-nickases fused to base editors. The original method is based on detecting double strand breaks in vitro. Here the authors modified Digenome-seq to identify genome-wide off-targets detecting single strand breaks of a C-to-U base editor fused to catalytically inactive dCpf1 (dCpf1-BE) in vitro. Genome-wide C-to-U conversion are then identified by a unique alignment pattern observed by whole genome sequencing (WGS). More genome-wide off-targets are observed for dCpf1-BE when compared to that of Cpf1 nuclease. Off-targets identified in vitro are then tested using targeted sequencing in HEK293 cells showing that the base editor and nuclease recognize different off-target sites.

Overall, this is a good study expanding on the previously developed digenome-seq method comparing off-target differences between the dCpf1 base editor and Cpf1 nuclease. Although the experimental work is convincing and worth publishing, the manuscript in its current form can be confusing in places and lacks depth in data description. Data need to be presented more clearly. In addition the discussion should be expanded to put findings in this study in perspective with literature on off-targets by nCas9-base editors using SNV analysis.

Major comments:

The manuscript lacks description of statistical analysis used. For example, are three-fold differences in off-target activity between dCpf1-BE and Cpf1 statistically significant. Or were the authors considering significant changes only when there was also a three-fold difference. Statistics used should be added to figure legends and method section. This applies to several figures, eg. Fig1, Sup Fig2, etc...

All analysis of modified Digenome-seq is focused on sites that have some resemblance with the target region. This does minimize the power of this whole genome method especially since genome-wide off-target effects have been reported for BE3 that DO NOT correlate with similarity to the target site (PMID:30819928). By focusing on sequencing similarity, the number of SSB (single strand breaks) observed for DYRK1A gets reduced from 123,309 SSBs to ~20 potential off-targets. This does not sound like an unbiased genome-wide approach. I feel strongly that if Digenome-seq is a good unbiased method, there should be more information that can be extracted from 123,309 SSBs. Focused analysis on sequence similarity should only be a follow-up analysis.

Are Digenome-seq results from a single replicate? This needs to be clearly stated. Would the most of the 123,309 SSBs be the same in biological replicates? Would they then be relevant? Biological replicates would be the best way to determine "real" off-targets. Did I miss that information?

I understand that there could be limitation to this method as it is performed in vitro. Possible advantages and limitations of Digenome-seq should be addressed in the discussion of the manuscript and compared to findings of relevant literature (PMID:30819928, PMID:30995674, ...)

Paragraph "Genome-wide off-target sites of dLbCpf1-BE revealed by Digenome-seq"

This paragraph needs a lot more detail. The numbers do not make any sense. I cannot evaluate this paragraph in its current form. How many genes were tested? Where do 106 target sites come from? I tried to do the math, but still cannot figure it out. Please add all information to the results section in the main text.

Line 151: "across three other target sites (Fig 3a)". This is not correct.

Line 151: "8 out of 59 cleavage sites". Where do they come from? Which genes? List # of sites per target gene

Fig3c: The sequence logo was determined for off-targets, but is not described in results section. Sequence motif analysis should also be shown for DYRK1A in Fig3c and for CDKN2A and EMX1 in Supp Fig 3. Findings should be summarized in the results section and interpretation added to the discussion.

Fig 3d/e: Please add base editing efficiency of dCpf1-BE in an additional column in the table. It is hard to compare between table and bar graph.

The results of Cpf1 variants tested in this manuscript are disappointing. However, the description of variants is vague and should be described in more detail. A comparison to other tested Cpf1 variants (PMID:30742127, PMID:30995674) would be helpful. Were any of the mutations similar or in similar regions?

Minor comments:

Line 27: Please specify range of SSBs rather than an average of 12 SSBs

Line 61: Please state clearly at beginning of first paragraph that targeted amplicon sequencing was used to determine indels

Line 62: It is hard to follow what the different targets without looking at the Supplemental Fig 1a. It would be helpful to take the time to list the nine target sites in the results paragraph.

Line 62: human genomic needs space

Suppl Fig 1 and Fig 2: What is the difference between DNMT1, EMX1 and DNMT1-3 and EMX1-2? Differences re-occur eg in Fig 3. Be consistent if it is they are the same or explain if they are different. Please adjust all affected nomenclature in text and figures.

Point-by-point Response

We would like to thank two anonymous reviewers for helpful comments. We addressed various issues raised by the reviewers as shown below and highlighted textual changes in our revised manuscript for ease of tracking.

Reviewers' comments:

Reviewer #1 (Remarks to the Author):

In the present study, Kim and colleagues intended to assess specificity of dCpf1 base editor, an alternative deamination-mediated base editing system that requires distinct PAM sequence and does not involve DNA nicking. By modifying their Digenome-seq method, which is based on *in vitro* digestion of isolated genome followed by WGS, the authors tried to provide a comprehensive assay to detect genome-wide deaminated sites. The authors argued that their method is highly sensitive and detects distinct off-target candidates from the ones of the nuclease version. However, as this method seems to detect many false positives and depends on the mismatch criteria to narrow down the number of candidate sites, the reviewer is skeptical of how much their modified Digenome-seq contributes to improve the detection of unexpected off-targets. Besides, the present study cannot address the recent issue of gRNA-independent off-targets for cytosine base editors. The findings that the off-targets for dCpf1 base editor are partly distinct from Cpf1, and that the Apobec1 variants reduces gRNA-dependent off-target, are not quite surprising. Therefore, this reviewer cannot recommend this manuscript to be published in Nature Communications.

Major comments

1. The authors need to mention and estimate the false positive rate of their method and perform comparison with pure prediction-based detection. It will be also beneficial to score off-target candidates based on their Digenome-seq result and compare them with actual off-target rates in the cells.

We have added the following sentences on p. 5: "Conversely, we were unable to detect *in vitro*-generated SSB sites that were homologous with the *DYRK1A* site when we used untreated genomic DNA, suggesting a very low number of false positive off-target sites."

Among the 106 candidate sites identified by Digenome-seq, 29 (=27%) were validated via targeted deep sequencing. However, it is possible that there are off-target sites with mutation frequencies below the next-generation sequencing detection limit (around 0.01-1%).

For comparison with pure prediction-based detection, we have now added the following sentences on p. 8: "The average number of potential off-target sites, defined as those sites with a PAM sequence (5'-TTTN-3') and having eight or fewer mismatches compared to the on-target site, was 247500 ± 42200 , making it practically impossible to validate them in cells (Supplementary Table 6). However, on average, dLbCpf1-BE- and USER-mediated Digenome-seq identified 12 *in vitro* cleavage sites (including the on-target site) per crRNA, a

testable number.”

It would be beneficial if we could predict off-target activity in cells from Digenome-seq results, but unfortunately there is a weak correlation between activity in cells and *in vitro*. We have mentioned this point in the Results section, as follows: “We also found that the ratio of the count of sequence reads with the same 5' end to the read depth in *in vitro* Digenome-seq results at validated dLbCpf1-BE off-target sites was poorly correlated with mutation frequencies in HEK293T cells ($R^2 = 0.30$) (Supplementary Fig. 6).”

2. If economically and technically acceptable, false positives could be reduced by increasing the depth of the reads, using other NGS system with longer reads, or enrichment of SSB samples by devising size selection (i.e. producing larger fragments at shearing, followed by end repair and size selection for smaller fragments).

It would be powerful if one could enrich SSB sites but we believe this area is beyond the scope of this study.

3. The authors need to assess if USER treatment alone may cause significant SSB.

Since the USER enzyme induces cleavage only at sites containing uracil, this enzyme can't induce cleavage without dCpf1-BE. Our modified Digenome-seq method captured SSB sites with homology with the on-target sequence, suggesting that it is very unlikely that the USER enzyme alone could yield SSB sites with high sequence homology even if it causes nonspecific cleavage.

4. The readers would be more interested if the authors could discuss if dCpf1 base editor is superior in its specificity, compared to conventional SpCas9-based one.

We have now added the following sentence on p. 7: “In addition, we compared the OTI (off-target effect index), the ratio of the sum of the mutation frequencies at validated off-target sites to the mutation frequency at the on-target site, of dLbCpf1-BE and BE3 (APOBEC1–Cas9 D10A nickase–UGI). We found that the OTI of dLbCpf1-BE (0.27 ± 0.16) was less than that of BE3 (0.45 ± 0.25), suggesting that the specificity of dLbCpf1-BE is higher than that of BE3 (Supplementary Table 5).”

Minor comments

L128-129: “A total of 123,309 SSB sites were captured at the DYRK1A site.”

Is this supposed to mean in the entire genome? Even so, this huge number suggests that this assay captures too much false positives probably due to technical and statistical limitation. The filtering based on the mismatch criteria cannot solve this statistical problem and thus the false positives remain.

Various unbiased off-target detection methods including CIRCLE-seq, GUIDE-seq, IDLVs, and HTGTS [PMID: 28459458, 25513782, 25599175, 25599175] use mismatch filtering because otherwise there are too many false positives. The original Digenome-seq method (Digenome 1.0) for identifying DSB sites is exceptional in that it does not require mismatch filtering. However, in this study, it was inevitable that mismatch filtering would need to be applied for Digenome-seq to detect SSBs generated *in vitro*. In addition, mismatch filtering has already been applied in the Digenome 2.0 analysis in which Cas9-cytosine base editor and -adenine base editor off-target effects were successfully detected (PMID: 28398345, 28398345).

.L143-144: "Note that the average number of *in vitro* LbCpf1 cleavage sites was 6 ± 3 , which is fewer than that of dLbCpf1-BE."

As the detection method for LbCpf1 nuclease off-target is different and perhaps more accurate, it may not be fair to compare these numbers without considering false positives.

We have now changed the sentence as follows: "Note that although a different detection method was used with dLbCpf1-BE, the average number of *in vitro* LbCpf1 cleavage sites was 6 ± 3 , which is fewer than that of dLbCpf1-BE."

L164-165: "The 20 validated off-target sites exhibited base editing frequencies that ranged from 0.1% to 22.4%."

Considering the detection limit of amplicon-seq by the NGS system, is it really possible to detect 0.1% SNV with statistical significance?

Background noise in targeted deep sequencing may vary depending on the cell line, the genomic position, etc. For this reason, we always include an 'untreated sample' as a control and compare its results with those of the treated sample during off-target validation steps in the human cell line. The base editing frequency of dLbCpf1-BE at validated sites was significantly higher than the background value of the untreated sample in targeted deep sequencing, as determined by statistical analysis ($P < 0.05$).

Reviewer #2 (Remarks to the Author):

The manuscript "Genome-wide specificity of dCpf1 cytidine base editors" describes off-target activity of dCpf1-BE (C-to-U base editor fused to catalytically inactive dCpf1) in comparison to Cpf1 nuclease. Targeted amplicon sequencing reveals that the dCpf1 base editor is more tolerant to mismatches than the Cpf1 nuclease. The authors describe a variation of Digenome-seq, a method that has been developed in the same lab and used to identify genome-wide off targets of RNA-guided nucleases (Cas9 and Cpf1) as well as Cas9-nickases fused to base editors. The original method is based on detecting double strand breaks *in vitro*. Here the authors modified Digenome-seq to identify genome-wide off-targets detecting single strand breaks of a C-to-U base editor fused to catalytically inactive dCpf1 (dCpf1-BE) *in vitro*. Genome-wide C-to-U conversion are then identified by a unique alignment pattern observed by whole genome sequencing (WGS). More genome-wide off-targets are observed for dCpf1-BE when compared to that of Cpf1 nuclease. Off-targets identified *in vitro* are then tested using targeted sequencing in HEK293 cells showing that the base editor and nuclease recognize different off-target sites.

Overall, this is a good study expanding on the previously developed digenome-seq method comparing off-target differences between the dCpf1 base editor and Cpf1 nuclease. Although the experimental work is convincing and worth publishing, the manuscript in its current form can be confusing in places and lacks depth in data description. Data need to be presented more clearly. In addition the discussion should be expanded to put findings in this study in perspective with literature on off-targets by nCas9-base editors using SNV analysis.

Major comments:

The manuscript lacks description of statistical analysis used. For example, are three-fold differences in off-target activity between dCpf1-BE and Cpf1 statistically significant. Or were the authors considering significant changes only when there was also a three-fold difference. Statistics used should be added to figure legends and method section. This applies to several figures, eg. Fig1, Sup Fig2, etc...

Apart from statistical methods such as commonly used p-values, the colored asterisks are used to indicate differences greater than three-fold in order to highlight dissimilarities in the patterns of dCpf1-BE and Cpf1 activity. This special notation is only applied in Fig. 1 and Supplementary Fig. 2. We have now described this notation in the Statistical analyses section in the Methods.

All analysis of modified Digenome-seq is focused on sites that have some resemblance with the target region. This does minimize the power of this whole genome method especially since genome-wide off-target effects have been reported for BE3 that DO NOT correlate with similarity to the target site (PMID:30819928). By focusing on sequencing similarity, the number of SSB (single strand breaks) observed for DYRK1A gets reduced from 123,309 SSBs to ~20 potential off-targets. This does not sound like an unbiased genome-wide approach. I feel strongly that if Digenome-seq is a good unbiased method, there should be more information that can be extracted from 123,309 SSBs. Focused analysis on sequence similarity should only be a follow-up analysis.

Various unbiased off-target detection methods including CIRCLE-seq, GUIDE-seq, IDLVs, and HTGTS [PMID: 28459458, 25513782, 25599175, 25599175] use mismatch filtering because otherwise there are too many false positive sites. The original Digenome-seq method (Digenome 1.0) for identifying DSB sites is exceptional in that it does not require mismatch filtering. However, in this study, it was inevitable that mismatch filtering would need to be applied for Digenome-seq to detect SSBs generated *in vitro*. In addition, mismatch filtering has already been applied in the Digenome 2.0 analysis in which Cas9-cytosine base editor and -adenine base editor off-target effects were successfully detected (PMID: 28398345, 28398345).

CRISPR-target independent off-target SNVs are already well covered in a previous study (PMID: 30819928). The off-target SNVs found by GOT1 do not overlap between samples and were unpredictable in each experiment. In this paper, we focus instead on dLbCpf1-BE target dependent off-target sites, which had not yet been analyzed. Searching for off-target candidates using only target sequence similarity means that actual off-target sites must be identified from a long list of candidates; in contrast, the Digenome-seq method, which combines sequence similarity with an *in vitro* detection method, can generate a list of the most likely off-target sites. To better illustrate this advantage, we have added “The average number of potential off-target sites, defined as those sites with a PAM sequence (5'-TTTN-3') and having eight or fewer mismatches compared to the on-target site, was 247500 ± 42200 , making it practically impossible to validate them in cells (Supplementary Table 6). However, on average, dLbCpf1-BE- and USER-mediated Digenome-seq identified 12 *in vitro* cleavage sites (including the on-target site) per crRNA, a testable number.” in the Discussion section.

Are Digenome-seq results from a single replicate? This needs to be clearly stated. Would the most of the 123,309 SSBs be the same in biological replicates? Would they then be relevant? Biological replicates would be the best way to determine “real” off-targets. Did I

miss that information?

We have added the following sentences on p. 6: “To check the reproducibility of Digenome-seq, we independently performed Digenome-seq using the *DYRK1A*-targeted dLbCpf1-BE twice, and found that the same 17 sites including the on-target site were captured by the two different experiments (Supplementary Fig. 3 and Supplementary Table 1). These experiments established that we could identify dLbCpf1-BE off-target sites using Digenome-seq with high reproducibility.”

I understand that there could be limitation to this method as it is performed *in vitro*. Possible advantages and limitations of Digenome-seq should be addressed in the discussion of the manuscript and compared to findings of relevant literature (PMID:30819928, PMID:30995674, ...)

We have now added the following sentences on p. 9: “The average number of potential off-target sites, defined as those sites with a PAM sequence (5'-TTTN-3') and having eight or fewer mismatches compared to the on-target site, was 247500 ± 42200 , making it practically impossible to validate them in cells (Supplementary Table 6). However, on average, dLbCpf1-BE- and USER-mediated Digenome-seq identified 12 *in vitro* cleavage sites (including the on-target site) per crRNA, a testable number.”, and “In previous studies, base editors consisting of Cas9 nickase and APOBEC1 induced not only gRNA-dependent off-target mutations but also gRNA-independent DNA or RNA mutations. Theoretically, dLbCpf1-BE could likewise induce crRNA-independent DNA or RNA editing in cells. To fully understand the off-target effects of dLbCpf1-BE, further studies are needed to identify crRNA-independent off-target effects.”

Paragraph “Genome-wide off-target sites of dLbCpf1-BE revealed by Digenome-seq”

This paragraph needs a lot more detail. The numbers do not make any sense. I cannot evaluate this paragraph in its current form. How many genes were tested? Where do 106 target sites come from? I tried to do the math, but still cannot figure it out. Please add all information to the results section in the main text.

Line 151: “across three other target sites (Fig 3a)”. This is not correct.

We have now changed the sentence as follows: “In addition, we also compared the *in vitro* cleavage sites induced by LbCpf1 nuclease and dLbCpf1-BE targeted to three additional target sites (*DNMT1*, *CKDN1A*, and *EMX1*) (Fig. 3b and Supplementary Fig. 4).”

Line 151: “8 out of 59 cleavage sites”. Where do they come from? Which genes? List # of sites per target gene

To clarify, we have now changed the sentence as follows: “ Only 9 out of 73 (=12%) of the cleavage sites identified using dLbCpf1-BE targeting *DYRK1A*, *DNMT1*, *CDKN1A*, and *EMX1* were also identified using LbCpf1 nucleases targeting the same sites (Fig. 3a, b and Supplementary Fig. 4).”

Fig3c: The sequence logo was determined for off-targets, but is not described in results section. Sequence motif analysis should also be shown for *DYRK1A* in Fig3c and for *CDKN2A* and *EMX1* in Supp Fig 3. Findings should be summarized in the results section and interpretation added to the discussion.

We have now added the following sentences on p. 6: “A sequence logo, generated from the DNA sequences at *in vitro* LbCpf1 nuclease cleavage sites, indicated that all nucleotide

positions contributed to specificity (Fig. 3c). However, a sequence logo generated from the dLbCpf1-BE-modified sites showed that the sequence of the PAM distal region was less important than the seed sequence for specificity (Fig. 3c)."

Fig 3d/e: Please add base editing efficiency of dCpf1-BE in an additional column in the table. It is hard to compare between table and bar graph.

We have now added Supplementary Figure 5.

The results of Cpf1 variants tested in this manuscript are disappointing. However, the description of variants is vague and should be described in more detail. A comparison to other tested Cpf1 variants (PMID:30742127, PMID:30995674) would be helpful. Were any of the mutations similar or in similar regions?

We have already employed the high-fidelity version of AsCpf1 reported in PMID:30742127 to improve dLbCpf1-BE. The N260A mutation in dLbCpf1-BE, which corresponds to the N282A mutation in AsCpf1 (HF1 version), had no significant effect on specificity. We added a detailed description of each mutation to the Methods section as follows: "incorporate mutations in the LbCpf1 domain (N256A, N260A, S348A, K514A, K881A, or K897A; these mutations respectively correspond to N278A, N282A, S376A, K523A, K949A, or K965A in AsCpf1)".

Minor comments:

Line 27: Please specify range of SSBs rather than an average of 12 SSBs

The number of off-target sites for each target site is shown in Supplementary Table 1.

Line 61: Please state clearly at beginning of first paragraph that targeted amplicon sequencing was used to determine indels

We have now changed the sentence as follows: "We first compared the insertion and deletion (indel) frequencies of LbCpf1 nuclease and the base editing frequency of dLbCpf1-BE at nine human genomic target sites (*CDKN2A*, *RUNX*, *FANCF*, *EMX1*, *DNMT1*, *LINC01551*, *DYRK1A*, *BCL2L13*, and *CLIC4*) in HEK293T cells using targeted deep sequencing."

Line 62: It is hard to follow what the different targets without looking at the Supplemental Fig 1a. It would be helpful to take the time to list the nine target sites in the results paragraph.

We have now changed the sentence as follows: "We first compared the insertion and deletion (indel) frequencies of LbCpf1 nuclease and the base editing frequency of dLbCpf1-BE at nine human genomic target sites (*CDKN2A*, *RUNX*, *FANCF*, *EMX1*, *DNMT1*, *LINC01551*, *DYRK1A*, *BCL2L13*, and *CLIC4*) in HEK293T cells using targeted deep sequencing."

Line 62: human genomic needs space

We added the sequence and genomic location of the nine target sites to Supplementary Fig. 1a.

Suppl Fig 1 and Fig 2: What is the difference between DNMT1, EMX1 and DNMT1-3 and

EMX1-2? Differences re-occur eg in Fig 3. Be consistent if it is they are the same or explain if they are different. Please adjust all affected nomenclature in text and figures.

The names of the target genes and crRNAs were adjusted so that they are uniform in the text and figures.

Reviewers' Comments:

Reviewer #1:

Remarks to the Author:

As had been pointed out by the both reviewers, the present method largely depends on mismatch filtering. In the authors responses, they argued that many of the previous methods also did the same way, and the sequence-independent off-target study should be done elsewhere. This argument does not gain novelty nor superiority of this study. The authors also argued that their Digenome-seq identified testable number of candidates. However, it seems that the mismatch filtering parameters were set in such a way that the number should be reduced to a good size in combination with their Digenome-seq results. Although this reviewer appreciates uniqueness of their method, it is hard to say "unbiased" and "genome wide" at the current standard of this field.

Reviewer #2:

Remarks to the Author:

The authors have answered all questions and demonstrated reproducibility of their method. The author have demonstrated the use of this in-vitro method to look at dCpf1-guided base editors. This is an alternative to SNV analysis performed by other groups. dCpf-independent off-targets could be detected by SNV analysis, but not by the method described in this manuscript. The authors could perform SNV analysis on the whole genome data sets used in this study for comparison and discussion.

Point-by-point response

We would like to thank two anonymous reviewers for helpful comments. We addressed issues raised by the reviewers as shown below.

REVIEWER COMMENTS

Reviewer #1 (Remarks to the Author):

As had been pointed out by the both reviewers, the present method largely depends on mismatch filtering. In the authors responses, they argued that many of the previous methods also did the same way, and the sequence-independent off-target study should be done elsewhere. This argument does not gain novelty nor superiority of this study. The authors also argued that their Digenome-seq identified testable number of candidates. However, it seems that the mismatch filtering parameters were set in such a way that the number should be reduced to a good size in combination with their Digenome-seq results. Although this reviewer appreciates uniqueness of their method, it is hard to say "unbiased" and "genome wide" at the current standard of this field.

Please note that all of the "unbiased" and "genome-wide" methods, including GUIDE-seq, DISCOVER-seq, BLISS, BLESS, CIRCLE-seq, SITE-seq, and HTGTS, except our original Digenome-seq, rely on sequence homology search or mismatch filtering to identify CRISPR off-target sites. Our original Digenome-seq is the only method that does not rely on mismatch filtering. But these methods including our original Digenome-seq identify DSB sites and cannot detect SSB or nick sites. To find nick patterns across the genome, we modified Digenome-seq and used mismatch filtering in this manuscript. Our method described in this manuscript is the only method developed thus far that allows identification of dCpf1 base editor off-target sites in the genome. We will consult with the editor whether we need to remove the term "unbiased" in our manuscript.

Reviewer #2 (Remarks to the Author):

The authors have answered all questions and demonstrated reproducibility of their method. The author have demonstrated the use of this in-vitro method to look at dCpf1-guided base editors. This is an alternative to SNV analysis performed by other groups. dCpf-independent off-targets could be detected by SNV analysis, but not by the method described in this manuscript. The authors could perform SNV analysis on the whole genome data sets used in this study for comparison and discussion.

Our method cannot identify guide RNA-independent off-target sites via SNV analysis, because we use USER (Uracil Specific Excision Reagent) to remove uracil in DNA prior to whole genome sequencing. Note that we identify guide RNA-dependent off-target sites with C-to-U conversion by monitoring cleavage (nick) patterns rather than detecting uracil directly.

Besides, it has already been shown that Cas9 base editors induce guide RNA-independent off-target mutations randomly across the genome. dCpf1 base editors are likely to induce such off-target mutations but it is way beyond the scope of our manuscript, which focuses on the guide RNA-dependent off-target activity of dCpf1 base editors.

Please note that we have already mentioned this issue in our manuscript on p. 9: "In previous studies, base editors consisting of Cas9 nickase and APOBEC1 induced not only gRNA-dependent off-target mutations but also gRNA-independent DNA or RNA mutations. Theoretically, dLbCpf1-BE could

likewise induce crRNA-independent DNA or RNA editing in cells. To fully understand the off-target effects of dLbCpf1-BE, further studies are needed to identify crRNA-independent off-target effects."

Reviewers' Comments:

Reviewer #1 (Remarks to the Author)

The weakness of this study is an arbitrary (or not explained) parameter setting for mismatch filtering and the lack of quantitative correlation with in vivo results, which raise concerns of both incomprehensiveness and false positives. For "unbiased" and "genome wide" studies, the readers today would expect more comprehensive analyses that include gRNA-independent off-targets or SNV. If these points are beyond the scope of this study, usability of the method would be limited, even though this reviewer acknowledges its uniqueness, and the difficult situation to perform additional experiments.

Reviewer #2 (Remarks to the Author)

The authors have demonstrated a creative off-target detection method that is worthy of publishing and of great interest to the community. The only thing this reviewer asks is that the authors put their work in context of published SNV off-target analysis in the discussion. How is this in-vitro manipulation followed by WGS better or a good alternative to simply performing WGS (without additional manipulation) and determining SNVs? I think this is a fair question and an important one to inform other scientists that want to perform similar experiments.

Point-by-point response

We would like to thank two anonymous reviewers for helpful comments. We addressed issues raised by the reviewers as shown below.

REVIEWERS' COMMENTS:

Reviewer #1 (Remarks to the Author):

The weakness of this study is an arbitrary (or not explained) parameter setting for mismatch filtering and the lack of quantitative correlation with in vivo results, which raise concerns of both incomprehensiveness and false positives. For “unbiased” and “genome wide” studies, the readers today would expect more comprehensive analyses that include gRNA-independent off-targets or SNV. If these points are beyond the scope of this study, usability of the method would be limited, even though this reviewer acknowledges its uniqueness, and the difficult situation to perform additional experiments.

To clarify parameter setting for mismatch filtering, we have added Supplementary Table 1. We chose the mismatch number of 8 as the cutoff value because WGS data obtained with intact genomic DNA did not show any false-positive sites with 8 or fewer mismatches.

Supplementary Table 1. The number of off-target candidates depending on the number of mismatches.

DYRK1A											
Number of mismatches at the protospacer sequence	0	1	2	3	4	5	6	7	8	9	10
Untreated	0	0	0	0	0	0	0	0	0	24	82
(+) dLbCpf1-BE/USER	1	0	0	0	1	2	4	1	4	27	109

Reviewer #2 (Remarks to the Author):

The authors have demonstrated a creative off-target detection method that is worthy of publishing and of great interest to the community. The only thing this reviewer asks is that the authors put their work in context of published SNV off-target analysis in the discussion. How is this in-vitro manipulation followed by WGS better or a good alternative to simply performing WGS (without additional manipulation) and determining SNVs? I think this is a fair question and an important one to inform other scientists that want to perform similar experiments.

We believe that we have already discussed this issue in the Discussion section. Our method is designed to capture guide RNA-dependent off-target sites and cannot identify guide RNA-independent off-target sites.